# Correlation between Apnea Severity and Sagittal Cephalometric Features in a Population of Patients with Polysomnographically Diagnosed Obstructive Sleep Apnea

**DOI:** 10.3390/jcm11154572

**Published:** 2022-08-05

**Authors:** Matteo Pollis, Frank Lobbezoo, Ghizlane Aarab, Marco Ferrari, Rosario Marchese-Ragona, Daniele Manfredini

**Affiliations:** 1Department of Medical Biotechnology, School of Dentistry, University of Siena, 53100 Siena, Italy; 2Department of Orofacial Pain and Dysfunction, Academic Centre of Dentistry Amsterdam (ACTA), University of Amsterdam and Vrije Universiteit Amsterdam, 1081 LA Amsterdam, The Netherlands; 3Department of Neurosciences, Otolaryngology Section, University of Padova, Via Belzoni, 160, 35121 Padova, Italy

**Keywords:** obstructive sleep apnea, cephalometry, craniofacial morphology

## Abstract

Background and Objective: Obstructive sleep apnea (OSA) is a sleep-related breathing disorder featuring a repeated closure of the upper airway during sleep. Craniofacial anatomy is a potential risk and worsening factor for OSA. This study aims to assess the relationship between cephalometric features of craniofacial morphology and OSA severity in a population of patients with OSA. Material and Methods: A sample of forty-two patients (*n* = 42, M = 76%, mean age = 57.8 ± 10.8) with a polysomnographically (PSG) confirmed diagnosis of OSA were recruited and underwent cephalometric evaluation of 16 cephalometric variables. In addition, the apnea–hypopnea index (AHI), oxygen desaturation (SatMin), Epworth sleepiness scale (ESS), and body mass index (BMI) were assessed. Then *t*-tests were performed to compare the values of all cephalometric variables between two AHI severity-based groups (mild-to-moderate = AHI ≤ 30; severe = AHI > 30). Single- and multiple-variable regression analyses were performed to assess the associations between AHI scores and cephalometric features. Results: Mean AHI, SatMin, and BMI were 31.4 ev/h, 78.7%, and 28.1, respectively. The cephalometric variables were not significantly different between the two OSA-severity groups (*p* > 0.05). Multiple-variable regression analyses showed that gonial angle and nasopharynx space were negatively associated with AHI, explaining 24.6% of the total variance. Conclusion: This investigation reported that severity of AHI scores in patients with OSA showed a negative correlation with gonial angle and nasopharynx space. As a general remark, although maxillofacial anatomy can be a predisposing factor for OSA, disease severity depends mainly upon other variables.

## 1. Introduction

Obstructive sleep apnea (OSA) is a sleep-related breathing disorder featuring a repeated collapse of the upper airway, often resulting in oxygen desaturation and arousals from sleep [1]. The collapse events can cause a partial (i.e., hypopnea event) or total closure (i.e., apnea event) of the upper airway [2,3,4]. According to a recent review, nearly 1 billion adults aged 30–69 years worldwide may have OSA, with an increasing prevalence over the past several decades [5]. The pathophysiology of OSA is complex, but it is suggested that anatomical features associated with a small upper airway volume (e.g., retrognathia, maxillary hypoplasia, tonsillar and lingual hypertrophy, excess adipose tissue around the airway lumen) may facilitate collapse during sleep when muscle compensation is absent [6,7,8,9,10].

Moderate-to-severe OSA is a risk factor for stroke and predisposes to a number of chronic cardiovascular, neurological, and metabolic diseases [11]. In addition, excessive daytime sleepiness in patients with OSAmay cause motor vehicle and work-related accidents. Due to its relevance as a condition that may have negative health consequences, OSA is the target of ongoing studies that enlightened several associated comorbidities and risk factors (e.g., obesity, older age, male gender, heredity, smoking, diabetes) [3,12,13,14].

The suspicion of OSA is based on the presence of clinical symptoms, such as excessive tiredness, daytime sleepiness, nocturia, loud snoring, witnessed apneas, and choking or gasping at night [15]. Definite diagnosis is made with polysomnography (PSG), which allows for measurement of the number of apnea and hypopnea events to rate the condition’s severity (apnea–hypopnea index [AHI]) [16]. However, PSG does not reveal the specific obstruction site, nor does it provide any differential diagnostic information as far as the potential relationship between the site of obstruction and OSA severity is concerned [15]. Thus, radiographic and endoscopic examinations have received attention as possible strategies to correlate PSG findings with the anatomy [17,18].

To this aim, lateral teleradiography of the head (TeleRx) is frequently adopted. It is a simple, reliable, and low-cost exam that provides a very limited radiation dose compared to computerized tomography (CT), and it enables an evaluation of facial anatomy by means of cephalometric analysis of hard tissues. Moreover, although TeleRx is a two-dimensional exam, it has sufficient accuracy for soft tissues to identify the possible sites of obstruction on a sagittal plane [19].

The correlation between OSA and craniofacial morphology is a much-debated issue in the literature [9,19,20,21,22,23,24,25]. In particular, although some studies report different features between patients with OSA and healthy individuals regarding soft tissues’ dimensions and volume (e.g., thicker soft palate, a major tongue volume, and a narrower cross-sectional area of the upper airway at the uvula level) [26,27,28], investigations on craniofacial anatomy provide conflicting results. A recent systematic review suggests the presence of an inferiorly positioned hyoid bone and a greater anterior facial height in patients with OSA compared to controls [20], but the strength of evidence is low due to the high heterogeneity of the reviewed studies [29,30,31]. Furthermore, it should be pointed out that most works compare patients with OSA with healthy individuals [24,28,30,32,33,34] without considering different degrees of OSA severity. Indeed, some studies assert that anatomical features like caudalization of hyoid bone and lower hypopharynx calibre are correlated with AHI scores [22,27]. Based on that, it can be also hypothesized that cephalometric features of face height and mandible divergence (e.g., sagittal facial projection, facial height and divergence) may condition the soft tissues around the upper airway [25,35].

Considering the above premises, the aim of the present study is to assess the relationship between cephalometric features and AHI severity in a population of patients with PSG-confirmed OSA. The outcome variables are 13 hard tissue-related variables concerning sagittal facial projection, facial height and divergence, mandible dimensions, and hyoid bone position, as well as three soft tissue-related variables concerning the nasopharynx space, the oropharynx space, and the soft palate length. The study hypothesis is that cephalometric outcome variables are correlated with AHI scores in a population of patients affected by OSA.

## 2. Materials and Methods

The study was prospectively performed at the Sleep Apnea Center of the University of Padova, Italy. A sample of 42 consecutive adult individuals (32 males; mean ± SD age = 57.8 ± 10.8 years) with a PSG-confirmed diagnosis of OSA (i.e., AHI over 5 events/h) took part to the study and underwent cephalometric evaluation. Exclusion criteria were a history of upper airway surgery (e.g., uvulopalatoplastic) or maxillofacial surgery (e.g., maxilla-mandibular advancement) for OSA [7], craniofacial syndromes (e.g., Treacher–Collins, Down, Pierre–Robin, Marfan), head–neck neoplasia, AHI lower than 5 events/h, and age <18 years. For each patient, OSA severity index (i.e., AHI), minimal oxygen desaturation (SatMin), Epworth sleepiness scale (ESS) [36], and body mass index (BMI) were assessed. The study protocol was approved by the University of Padova’s IRB in date 8 January 2016 as part of the routine activities of Sleep Apnea Center (approval code 080116/A), and each patient signed written consent to participate.

All patients underwent single-night PSG recordings in a sleep laboratory setting. According to American Academy of Sleep Medicine criteria [1], a >90% drop in the flow amplitude compared to pre-event baseline for at least 10 s was considered an apnea, and a ≥30% drop in the flow amplitude respect to pre-event baseline for at least 10 s associated with a minimal reduction of 3% of SatMin or an arousal was considered an episode of hypopnea [16]. Analyses of traces were performed by a certified sleep doctor (R.M.-R.). For each patient, lateral teleradiography was performed by experienced radiologist technicians. A cephalostat was used to keep the subject’s head in a natural head position.

Dolphin Imaging 11.8 (Dolphin Imaging and Management Solutions, Chatsworth, CA, USA) software was used by a certified orthodontist (D.M.) to assess 16 cephalometric variables [20]. Tweed’s cephalometric analysis and analysis of upper airway morphology were adopted.

The cephalometric points used to define the variables below are reported in Table 1.

The 16 variables were divided into two groups: hard tissue-related variables and soft tissue-related variables. The hard tissue-related variables (n = 13) include

-SNA, SNB, and ANB (sagittal facial projection);-Ar-Go⌃Me, Ar-Go/N-Me (%), S-N⌃Ar, Na-Me, MP⌃S-Na, and S-Go (facial height and divergence);-Go-Gn and Ar-Go (mandibular dimensions);-MP-Hy: perpendicular distance between mandibular plane and hyoid point; and-Hy-PAS: distance, parallel to mandibular plane, between hyoid point and posterior pharyngeal wall.

The soft tissue-related variables (n = 3) include

-VPAS (nasopharynx space): minimal distance, parallel to horizontal plane, between posterior pharyngeal wall and posterior soft palate surface;-PAS (oropharynx space): minimal distance, parallel to horizontal plane, between posterior pharyngeal wall and posterior lingual surface; and-SNP-P (soft palate length): distance between SNP and P point.

For each cephalometric variable, the mean ± SD value in the study sample was calculated. Single-variable regression analysis was performed to assess the correlation between cephalometric features and AHI. Variables that were significant at *p* < 0.15 were arbitrarily entered into a multiple-variable regression analysis, and adjusted for age, gender and BMI, to assess the level of variance in AHI scores that is explained by cephalometric variables. Furthermore, to further mine the data, cephalometric variables were compared between two AHI severity-based groups (mild-to-moderate = AHI ≤ 30; severe = AHI > 30) by means of *t*-tests, based on normal distribution of values as assessed with Levene’s test. The cut-off for statistical significance was set at *p* < 0.05, with Bonferroni correction for multiple testing, when needed. All statistical procedures were performed with the software SPSS 25.0 (IBM Software, Milan, Italy).

## 3. Results

Based on AHI index (cut-off = 30 events/h), 23 patients had a mild-to-moderate OSA and 19 a severe OSA. The mean AHI of the participants was 31.4 (± 19.8) events/h. The mean SatMin was 78.8% (± 8.1), the mean ESS was 7.0 (± 4.2), and the mean BMI was 28.1 (± 5.5) (i.e., mild overweight).

The *t*-tests showed that cephalometric variables were not significantly different between the two OSA-severity groups (*p* > 0.05) (Table 2).

Single-variable regression analysis showed that two cephalometric features were eligible to enter the multiple-variable analysis (i.e., VPAS, *p* = 0.033; Ar-Go⌃Me (°), *p* = 0.108) (Table 3). The two factors were entered in a backward conditional regression analysis, adjusted for age, gender and BMI, and were both significant at *p* < 0.05 in the final regression model (Table 4). The percentage of variance in the AHI score (Nagelkerke R-squared) that is explained by the two features was 24.6%.

## 4. Discusion

In recent years, OSA has gained an increasing interest within several medical communities, due to its remarkable prevalence [5] and important role as a risk factor for cardiovascular and metabolic disorders [11]. The pathophysiology of OSA is complex, with a number of anatomical and non-anatomical factors that may predispose to obstructive events. Anatomical factors may determine a predisposition to the reduction of pharyngeal airspace in multiple ways. For instance, they can facilitate airway collapse due to the surrounding fat tissue in obese patients and/or influence the muscle responsiveness to a respiratory arousal [6,13,19,20]. Anatomical factors include features of the soft and hard tissues, although the literature is not conclusive on their possible role in patients with OSA. In particular, although there seems to be an absence of differences between the facial skeleton patients with OSAand healthy individuals, the possible relationship between cephalometric features and the severity of disease in populations of patients with OSA, as identified by AHI scores, has never been assessed. Thus, the present study was carried out to get deeper into the relationship between cephalometric features and AHI scores in a population of PSG-confirmed patients with OSA. Results show no differences as for cephalometric features between the patients with mild-to-moderate OSA and those with severe OSA (*p* > 0.05). Multiple-variable regression analysis reveals that two cephalometric features, viz., nasopharynx space (VPAS, *p* = 0.007) and gonial angle (Ar-Go⌃Me (°), *p* = 0.021), are negatively correlated with AHI and explain about one-fourth of the variance (24.6%). This suggests that these cephalometric features might be potential predictors of predisposition to OSA. Hence, in spite of a general suggestion that OSA severity does not seem to depend directly on soft- or hard-tissue craniofacial features, they may be worthy of further exploration within the broader context of multifactorial OSA etiology. Such results are in line with those reported by Cillo et al. [23] and may be viewed as a support for the complexity of OSA physiopathology, which prevents one from identifying a clear-cut relationship with any given potential risk factor. These findings add to the discussion on the available literature on hard-tissue and soft-tissue features in patients with OSA.

Concerning hard-tissue anatomy, some works support the role of a hyperdivergent facial pattern as a risk factor for OSA, which is in contrast with our findings. A hyperdivergent facial pattern is generally characterized by a large sagittal and vertical facial discrepancy (i.e., ANB, MP⌃S-Na, MP-Hy and Na-Me increased), with a possible combination of retrognathic and clockwise-rotated mandible that may determine a narrowing of pharyngeal airspace [37]. Furthermore, Neelapu et al. reports an increase of anterior vertical facial height and a more caudal position of the hyoid bone in patients with OSA, both correlated with a sagittal mandibular retrusion [20]. Based on these findings, the authors postulate that patients with a hyperdivergent facial pattern would predispose one to upper airway obstruction. On the contrary, some other works support the association of OSA with a hypodivergent facial pattern, because of a shorter airway length and a greater proximity between soft palate, tonsils, tongue, and larynx [38]. Such findings are also reported by other authors and are in line with our findings [8,39,40]. In addition, a recent study assessing the relationship between craniofacial anatomy and severity of OSA reports a weak positive correlation between total cranial vertical rotation and AHI, which suggests that a reduced mandibular length may influence the severity of OSA [41]. After these considerations, it is likely that both facial biotypes potentially predispose one to the airway obstruction through different mechanisms. On the other hand, it cannot be excluded that different results of single papers are due to a single-variable approach to the study of a complex multifactorial phenomenon, which is a strategy that may lead to inconsistent findings across different study populations.

The role of soft tissues in OSA has been widely debated in the literature [13]. In a systematic review, Gottlieb et al. support the role of fat tissue in the lateral pharyngeal wall, soft palate length, and tongue volume as OSA predisposing factors [42]. Moreover, in line with our findings, Neelapu et al. reports a significant decrease in VPAS and PAS, which could depend on the encroachment and position of other structures [20]. However, though Lowe et al. considers soft palate size as an apnea severity indicator [43], recent reviews recommended caution in the interpretation of such features [19]. Importantly, only a few studies support a correlation with OSA severity, considering some anatomical features as predisposing to different OSA degrees. Costa–Suosa et al. found a negative association of VPAS and PAS with AHI and a positive association between AHI and distance between mandibular plane and the hyoid point (MP-Hy). A possible explanation is that a more caudal position of the hyoid bone determines a pharynx volume reduction and, consequently, an aggravation of OSA severity [22]. Other authors agree with these findings [10,19,24,27]. Such feature may contribute to sagittal airway obstruction, but some authors suggest that, on the contrary, it may be the consequence of the high pressure exerted on pharynx soft tissue [44]. Thus, in general, evidence about the correlation between soft tissue anatomy and OSA severity remains unclear. In light of this, it is important to remiember that soft tissues may also be influenced by muscle relaxation during sleep, whereas all cephalometric measurements are done during wakefulness. Correlation studies on the wakefulness vs. sleep measurements as well as more large-scale studies, taking into account multiple features of OSA physiopathology, are needed to get deeper into this topic.

In addition, an interesting and recent result about the relationship between craniofacial predisposing factors and subtype of OSA (i.e., positional OSA and REM-related OSA) is proposed by Jo et al. The authors report that REM-related patients with OSA have a lower severity of sleep apnea than non-REM-related patients and REM sleep dependency are associated with anatomical factors (i.e., lower PAS, shorter MP-Hy, shorter SNP-P) whereas patients with positional OSA do not show such a tendency [45].

Thus, despite the amount of literature on the correlation between OSA and craniofacial anatomy, results are often conflicting, and our investigation is no exception. The complexity of OSA pathophysiology may explain the difficulties formulating a simple take-home message for the dentists working in this field. Nonetheless, considering the increasing role of the orthodontist as a potential OSA sentinel and care provider in selected cases of patients who need a mandibular advancement device, it is recommended that this line of research be improved and developed within the concept of a multifaceted OSA generator model. As reported above, the most recent knowledge supports the role of soft tissues and several non-anatomical factors, such as low arousal threshold, high loop gain, and poor muscle responsiveness, as significant predisposing factors for OSA [13]. Despite this, hard tissue-related anatomy (e.g., micrognathia, retrognathia, inferior positioning of the hyoid bone) remains important as a contributing factor to OSA predisposition in some populations with other concurrent physical features [6,42,46].

Considering the limitations of this study, the Authors did not perform a prior sample size calculation, and the lack of strong significance in the results may be influenced by the small sample size and low power of the study. It must be remarked that the sample included all Caucasian patients affected by OSA. This means that an enlarged sample controlled for ethnicity and also including healthy controls might be useful to build more solid regression analysis models. Moreover, factors concerning the potential differences of cephalometric analyses with respect to measurements during sleep as well as the adoption of specific supine or lateral AHI must be considered. Therefore, in the future it would be desirable to set up standardized studies, mostly in the patient recruitment and the cephalometric measurements. Lateral teleradiography was performed in the upright position, in contrast to patients’ supine position during sleep, which can influence the results of this study. Considering this issue, it must also be borne in mind that cephalometric analysis evaluates only the sagittal dimension with a bidimensional assessment, without any relation to the frontal plane. Therefore, potential predisposing anatomical features (e.g., dome palate, transverse dimensions of the airway) cannot be investigated. Moreover, the pharynx has a tendency to laterally collapse, which can be detected only by means of second-step assessments, such as CT or endoscopy. Thus, more research focused on dynamic (e.g., endoscopy) and volumetric examinations should be conducted, especially considering supine AHI, to test the hypothesis of a correlation with craniofacial morphology in the ecological environment and conditions. Studies on the prediction of effectiveness of mandibular advancement devices for OSA reduction, based on the anatomy and endoscopic findings, may contribute to the design of such tailored research [35,39].

## 5. Conclusions

Within the limitations of this study, this investigation reported that the severity of AHI scores in patients with OSA not associated with specific craniofacial parameters (with the exception of a negative correlation with gonial angle and nasopharynx space) showed a negative correlation with the gonial angle and nasopharynx space (*p* < 0.05). As a general remark, although maxillofacial anatomy can be a predisposing factor for OSA, disease severity depends mainly upon other variables.

## Figures and Tables

**Table 1 jcm-11-04572-t001:** Cephalometric points considered in the study.

Point	Symbol	Definition
Sella	S	Mean point of sella turcica
Nasion	Na	Most anterior point of fronto-nasal suture
A Point	A	Deepest anterior point on maxilla anterior concavity
B Point	B	Deepest anterior point on mandibular symphysis
Porion	Po	Most superior point on external auditory meatus
Orbitale	Or	Most inferior point on lower border of the bony orbit
Pogonion	Pg	Most anterior point on mandibular symphysis
Pterion	Pt	Most posterior-superior point on pterygo-maxillary fissure
Basion	Ba	Most anterior-inferior point on foramen magnum
Articulare	Ar	Most posterior point on condyle neck
Gnation	Gn	Most anterior-inferior point on mandibular symphysis
Gonion	Go	Intersecting point between mandibular plane and tangent line to posterior mandibular border
Menton	Me	Most inferior point on mandibular symphysis
Anterior Nasal Spine	ANS	Most anterior point of hard palate
Posterior Nasal Spine	PNS	Most posterior point of hard palate
Condilo	Co	Most superior point on condyle head
Sigmoid Incision	Sg	Deepest point on sigmoid incision
Posterior Border of Mandibular Branch	-	Most posterior point on mandibular branch
Anterior Border of Mandibular Branch	-	Most anterior point on mandibular branch
Hyoid Point	Hy	Most anterior-superior point on hyoid bone
Uvula Apex	P	Inferior tip of uvula

**Table 2 jcm-11-04572-t002:** Comparison of mean and SD values of cephalometric variables and OSA indices between mild–moderate and severe OSA groups by means of *t*-tests.

Variables	Mild–Moderate OSA Group Means	Severe OSA Group Means	Sig. (2-Tailed)
	**(n = 23)**	**(n = 19)**	
Hard tissue-related variables			
SNA (°)	81.6 ± 4.0	80.6 ± 3.9	0.530
SNB (°)	83.7 ± 3.4	79.3 ± 4.0	0.318
ANB (°)	1.7 ± 3.4	1.3 ± 3.3	0.738
Ar-Go⌃Me (°)	123.9 ± 6.8	120.9 ± 6.7	0.160
Ar-Go/N-Me %	68. ± 5.1	70.0 ± 4.8	0.423
S-N⌃Ar (°)	124.4 ± 6.1	124.9 ± 5.7	0.788
Na-Me (mm)	119.1 ± 5.5	117.8 ± 6.9	0.473
MP⌃S-Na (°)	31.6 ± 6.9	29.9 ± 6.1	0.392
S-Go (mm)	81.9 ± 7.1	82.3 ± 5.8	0.854
Go-Gn (mm)	79.3 ± 6.7	78.1 ± 6.6	0.556
Ar-Go (mm)	51.7 ± 6.8	52.2 ± 4.7	0.784
Mp-Hy (mm)	21.1 ± 6.1	20.9 ± 4.8	0.342
Hy-PAS (mm)	32.4 ± 4.6	31.8 ± 5.2	0.131
Soft tissue-related variables			
VPAS (mm)	7.3 ± 2.6	6.6 ± 2.8	0.374
PAS (mm)	10.8 ± 4.6	10.4 ± 4.4	0.466
SNP-P (mm)	37.8 ± 4.5	37.0 ± 5.1	0.432

**Table 3 jcm-11-04572-t003:** Single-variable regression between AHI and cephalometric variables.

Variables	AHI				
	**Sig.**	β **Coefficient**		**Confidence Interval** **(95%)**	
			**Inferior**		**Superior**
Hard tissue-related variables					
SNA (°)	0.265	−0.879	−2.451		0.876
SNB (°)	0.372	−0.698	−2.259		0.694
ANB (°)	0.681	−0.459	−2.700		1.781
Ar-Go⌃Me (°)	0.108	−0.732	−1.632		0.168
Ar-Go/N-Me %	0.296	0.664	−0.602		1.930
S-N⌃Ar (°)	0.778	0.540	−1.294		0.876
Na-Me (mm)	0.627	0.251	−0.785		1.1286
MP⌃S-Na (°)	0.570	−0.275	−1.246		0.696
S-Go (mm)	0.189	0.633	−0.324		1.590
Go-Gn (mm)	0.545	−0.298	−1.288		0.691
Ar-Go (mm)	0.470	0.398	−0.704		1.499
Mp-Hy (mm)	0.269	0.630	−0.506		1.767
Hy-PAS (mm)	0.522	−0.530	−2.188		1.128
Soft tissue-related variables					
VPAS (mm)	0.033	−2.584	−4.946		−0.221
PAS (mm)	0.312	−0.806	−2.399		0.787
SNP-P (mm)	0.824	−0.150	−1.507		1.207

**Table 4 jcm-11-04572-t004:** Multiple-variable regression analysis. Factors included in the final model.

Variables			AHI			
	**Sig.**	β **Coefficient**		**Confidence Interval** **(95%)**		**R2**
				**Inferior**	**Superior**	
Ar-Go⌃Me (°)	0.021	−1.013		−1.865	−0.160	24.6%
VPAS (mm)	0.007	−3.223		−5.519	−0.928	

## Data Availability

The data that support the findings of this study are available from the corresponding author, [MP], upon reasonable request.

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
