# Peer review of "Correlation between Apnea Severity and Sagittal Cephalometric Features in a Population of Patients with Polysomnographically Diagnosed Obstructive Sleep Apnea"

_jcm, 2022, doi:10.3390/jcm11154572_

Round 1

Reviewer 1 Report

OSA is a progressive health and economic issue. Finding predictors which can help dental professionals to identify patient with an increased risk for OSA is crucial. It has been suggested for many years that anatomical features are correlated with OSA. Therefore, the authors aimed to investigate weather anatomical morphology – objectified with the use of cephalometry – was associated with OSA severity. 

I have to commend the authors for their efforts in this endeavor.

I do have some inquiries and general comments on the performed study:

-       In the M&M section the author indicates that for each patient lateral teleradiography was performed by experienced radiologist technicians. Different papers have illustrated that patient posture during the X-ray can alter the cephalometric measurements. The authors, did not specify if the X-rays were all performed in a standardized way in order to achieve comparable images and to minimize measurement errors. Could the authors specify and clarify weather this was the case? And if so, how?

-       In the M&M section the Authors did not specify which cephalometric analyses was used, i.e., Steiner analysis or Downs analysis?

-       There are papers reporting similar studies, using different cephalometric measurements. Why did the authors choose to use these cephalometric measurements parameters?

-       In the M&M section the authors state that a sample of 42 consecutive adult individuals with a PSG-confirmed diagnosis of OSA took part to the study. Can the authors elaborate more on why they settled for 42 individuals? This looks to be a quite random number?

-       Why did the authors not include a control group?

-       Recently, Jo, JH. et al. reported in a similar study that when looking at specific PSG findings, only REM sleep dependency was associated with anatomical factors. Did the authors evaluate their study population for these specific PSG findings? (Jo, JH., Kim, SH., Jang, JH. et al. Comparison of polysomnographic and cephalometric parameters based on positional and rapid eye movement sleep dependency in obstructive sleep apnea. Sci Rep 12, 9828 (2022). https://doi.org/10.1038/s41598-022-13850-6)

-       based on the 2D cephalometry technique used by the authors, a main limitation and point of concern of this study is the fact that the transversal anatomical features and dimension – which can also have an influence on OSA severity – are not investigated, i.e., dome palate, lateral pharyngeal dimensions, etc.

-       In the discussion section the authors rightfully state that that “the soft tissues may also be influenced by muscle relaxation during sleep, whilst all cephalometric measurements are done during wakefulness”. I would suggest to the authors to also note that teleradiography is performed in upright position which is on contrast to patient supine position during sleep, which of course also influences this study and it’s results.

-       In contrast to the authors, Tepedino et al. recently concluded that certain anatomical factors maybe correlated to obstructive sleep apnea severity. How do the authors explain this discrepancy? (Tepedino M, Illuzzi G, Laurenziello M, et al. Craniofacial morphology in patients with obstructive sleep apnea: cephalometric evaluation. Braz J Otorhinolaryngol. 2022;88(2):228-234. doi:10.1016/j.bjorl.2020.05.026)

-       Can the authors elaborate on why in the pediatric OSA literature there is more conclusive evidence on the correlation between anatomical factors and OSA (severity)?

-       In the discussion section, the authors state ”Despite that, hard tissue related anatomy (e.g., micrognathia, retrognathia, inferior positioning of the hyoid bone) remains essential as a contributing factor to OSA predisposition in some populations with other concurrent physical features”. Based on the study results maybe this sentence should be nuanced slightly.

-       In the conclusion section, I would suggest to the authors changes the first sentence to: Within the limitations of this study, this investigation showed that severity of AHI scores in patients with OSA is not associated with specific craniofacial parameters, with the exception of a negative correlation with gonial angle and nasopharynx space.

Author Response

Dear reviewer,

thank you for your questions. We have provided a point-by-point response as follow:

In the M&M section the author indicates that for each patient lateral teleradiography was performed by experienced radiologist technicians. Different papers have illustrated that patient posture during the X-ray can alter the cephalometric measurements. The authors, did not specify if the X-rays were all performed in a standardized way in order to achieve comparable images and to minimize measurement errors. Could the authors specify and clarify weather this was the case? And if so, how?

Lateral Teleradiography was performed keeping the subject’s head in their “natural head position”. This position is reproducible and maintains the physiologic curve of cervical spine. A cephalostat was used to performed the examination. It has been replacing the Frankfort plane-related position.

-        In the M&M section the Authors did not specify which cephalometric analyses was used, i.e., Steiner analysis or Downs analysis?

        The cephalometric analysis by Tweed was adopted.

-        There are papers reporting similar studies, using different cephalometric measurements. Why did the authors choose to use these cephalometric measurements parameters?

        We evaluated many previous studies with the same/similar aim(s) of our study (e.g., Neelapu et al. 2016, Costa e Sousa et al. 2013, Hoekema et al. 2003 Ozturk et al 2011 etc). After have considered the most adopted measurements among the studies we chose the ones compatible with Tweed’s analysis.

-        In the M&M section the authors state that a sample of 42 consecutive adult individuals with a PSG-confirmed diagnosis of OSA took part to the study. Can the authors elaborate more on why they settled for 42 individuals? This looks to be a quite random number?

The patients were recruited consecutively between the period of January 2016-January 2017

-       Why did the authors not include a control group?

This is a pertinent suggestion, since the inclusion of a control group would have surely enhanced the validity of our findings. Unfortunately, we have restrictions due to the National laws of radioprotection and we could not perform teleradiography in healthy individual.

-        Recently, Jo, JH. et al. reported in a similar study that when looking at specific PSG findings, only REM sleep dependency was associated with anatomical factors. Did the authors evaluate their study population for these specific PSG findings? (Jo, JH., Kim, SH., Jang, JH. et al. Comparison of polysomnographic and cephalometric parameters based on positional and rapid eye movement sleep dependency in obstructive sleep apnea. Sci Rep 12, 9828 (2022). https://doi.org/10.1038/s41598-022-13850-6).

        Since our study was completed before the publication of Jo et al. paper, we did not consider it in references. However, the conclusions are remarkable and we can add a paragraph in discussion section.

-      Based on the 2D cephalometry technique used by the authors, a main limitation and point of concern of this study is the fact that the transversal anatomical features and dimension – which can also have an influence on OSA severity – are not investigated, i.e., dome palate, lateral pharyngeal dimensions, etc.

        Right point. We have updated the limitation section focusing on this aspect.

-      In the discussion section the authors rightfully state that that “the soft tissues may also be influenced by muscle relaxation during sleep, whilst all cephalometric measurements are done during wakefulness”. I would suggest to the authors to also note that teleradiography is performed in upright position which is on contrast to patient supine position during sleep, which of course also influences this study and it’s results.

        Thank you for the point. We have updated the limitation section.

-        In contrast to the authors, Tepedino et al. recently concluded that certain anatomical factors maybe correlated to obstructive sleep apnea severity. How do the authors explain this discrepancy? (Tepedino M, Illuzzi G, Laurenziello M, et al. Craniofacial morphology in patients with obstructive sleep apnea: cephalometric evaluation. Braz J Otorhinolaryngol. 2022;88(2):228-234. doi:10.1016/j.bjorl.2020.05.026).

        Thank you for mention it. It is an interesting paper and one of the few works that reports a correlation with OSA severity. Some different cephalometric measurements and a wider sample size may explain the discrepancy.

-        Can the authors elaborate on why in the pediatric OSA literature there is more conclusive evidence on the correlation between anatomical factors and OSA (severity)?

       In this paper we considered only adult patients. We decided to not discuss pediatric OSA.

-      In the discussion section, the authors state ”Despite that, hard tissue related anatomy (e.g., micrognathia, retrognathia, inferior positioning of the hyoid bone) remains essential as a contributing factor to OSA predisposition in some populations with other concurrent physical features”. Based on the study results maybe this sentence should be nuanced slightly.

        Thank you for the advice. We have modified the sentence.

-      In the conclusion section, I would suggest to the authors changes the first sentence to: Within the limitations of this study, this investigation showed that severity of AHI scores in patients with OSA is not associated with specific craniofacial parameters, with the exception of a negative correlation with gonial angle and nasopharynx space.

        Thank you for the advice. We have updated the sentence.

Reviewer 2 Report

Discussion section can be improved with more correlation stated with previous studies. Figures not included in the manuscript. Please include.

Author Response

Dear reviewer,

thank you for your questions. We have provided a point-by-point response as follow:

  • Discussion section can be improved with more correlation stated with previous studies.

Thank you for your advice. We have improved the discussion section citing more recent studies.

  • Figures not included in the manuscript. Please include.

This is an interesting suggestion, even if is not easy to deal with. Please let us know which, if any, specific figures may help us increasing the quality of our work

Reviewer 3 Report

Dear authors,

Thanks for submitting your manuscript for review. I have the following comments:

Major
-please state weather a power analysis calculation was conducted before starting the research. This is important because some of your results may be under powered

-several cephalometric articles reach your same conclusion. Please state what set your article apart from previous published data. What is the most important take a message of your research

Minor:

-please review the grammar. There are several sentences that start with the Pastans and continue on the present and the other way around. In addition there are several colloquial terms for example “ Obstruction that occurs like a narrowing « 

- OAHI is a better term when discussing obstructive sleep apnea.

-patients with OSA is a better term compared to OSA patients

- reference when is appropriate. However, you are missing the AASM manual of scoring  please add this to the references 

Author Response

Dear reviewer,

thank you for your suggestion. We have provided a point-by-point response as follow:

Please state weather a power analysis calculation was conducted before starting the research. This is important because some of your results may be under powered.

This is an interesting suggestion. Due to the convenience sample we recruited for this study in the absence of a control group as well as the absence of a standard of reference outcome variable for this kind of investigations, power analysis was not performed. Based on your suggestion, we have added it as a limitation of the study. 

-several cephalometric articles reach your same conclusion. Please state what set your article apart from previous published data. What is the most important take a message of your research.

We have now modified the conclusions to highlight the results (negative correlation with gonial angle and pharynx space).

Minor:

-please review the grammar. There are several sentences that start with the Pastans and continue on the present and the other way around. In addition there are several colloquial terms for example “ Obstruction that occurs like a narrowing « 

Thank you for the advices. We have modified the tenses as suggested.

- OAHI is a better term when discussing obstructive sleep apnea.

We fully agree with your suggestion, but we are not sure it actually applies to our paper. Indeed, we calculated all apnea events.

-patients with OSA is a better term compared to OSA patients

Thank you for your advice. We have modified the text.

- reference when is appropriate. However, you are missing the AASM manual of scoring please add this to the references 

Thank you. We have updated the references.